# Early Onset Ataxia with Comorbid Dystonia: Clinical, Anatomical and Biological Pathway Analysis Expose Shared Pathophysiology

**DOI:** 10.3390/diagnostics10120997

**Published:** 2020-11-24

**Authors:** Deborah A. Sival, Martinica Garofalo, Rick Brandsma, Tom A. Bokkers, Marloes van den Berg, Tom J. de Koning, Marina A. J. Tijssen, Dineke S. Verbeek

**Affiliations:** 1Department of Paediatric Neurology, Beatrix Children’s Hospital, University Medical Center Groningen, University of Groningen, 9700 RB Groningen, The Netherlands; m.garofalo@student.rug.nl (M.G.); R.Brandsma-3@umcutrecht.nl (R.B.); tombokkers@live.nl (T.A.B.); m.van.den.berg.21@student.rug.nl (M.v.d.B.); 2Department of Neurology, Beatrix Children’s Hospital, University Medical Center Groningen, University of Groningen, 9700 RB Groningen, The Netherlands; t.j.de.koning@umcg.nl (T.J.d.K.); m.a.j.de.koning-tijssen@umcg.nl (M.A.J.T.); 3Department of Genetics, Beatrix Children’s Hospital, University Medical Center Groningen, University of Groningen, 9700 RB Groningen, The Netherlands; d.s.verbeek@umcg.nl

**Keywords:** clinical genetics, early onset ataxia, dystonia, neurodevelopment, network analysis, bioinformatics, ataxia, phenotype, child

## Abstract

In degenerative adult onset ataxia (AOA), dystonic comorbidity is attributed to one disease continuum. However, in early adult onset ataxia (EOA), the prevalence and pathogenesis of dystonic comorbidity (EOAD^+^), are still unclear. In 80 EOA-patients, we determined the EOAD^+^-prevalence in association with MRI-abnormalities. Subsequently, we explored underlying biological pathways by genetic network and functional enrichment analysis. We checked pathway-outcomes in specific EOAD^+^-genotypes by comparing results with non-specifically (in-silico-determined) shared genes in up-to-date EOA, AOA and dystonia gene panels (that could concurrently cause ataxia and dystonia). In the majority (65%) of EOA-patients, mild EOAD^+^-features concurred with extra-cerebellar MRI abnormalities (at pons and/or basal-ganglia and/or thalamus (*p* = 0.001)). Genetic network and functional enrichment analysis in EOAD^+^-genotypes indicated an association with organelle- and cellular-component organization (important for energy production and signal transduction). In non-specifically, in-silico-determined shared EOA, AOA and dystonia genes, pathways were enriched for Krebs-cycle and fatty acid/lipid-metabolic processes. In frequently occurring EOAD^+^-phenotypes, clinical, anatomical and biological pathway analyses reveal shared pathophysiology between ataxia and dystonia, associated with cellular energy metabolism and network signal transduction. Insight in the underlying pathophysiology of heterogeneous EOAD^+^-phenotype-genotype relationships supports the rationale for testing with complete, up-to-date movement disorder gene lists, instead of single EOA gene-panels.

## 1. Introduction

The diagnosis “early onset ataxia” refers to a group of rare, genetically inheritable diseases with an estimated prevalence of 14.6 per 100,000 individuals, initiated before the 25th year of life. These “ataxic syndromes” involve a heterogeneous group of underlying disorders that may phenotypically involve: (a) pure ataxic features; (b) predominant ataxic features in combination with other comorbid movement disorder features; (c) mild ataxic features in combination with other primary movement disorder features; (d) hardly discernible, disputable or even absent ataxic features, but with an underlying diagnosis that is phenotypically described as ataxic in the Online Mendelian Inheritance in Man (OMIM) database [1]. Depending on the age of the patient at disease presentation, patients are categorized as ‘early onset ataxia’ (EOA, i.e., initiation before 25 years of age) or degenerative ‘adult onset ataxia’ (AOA, i.e., initiation after 25 years of age) [2]. Both disease groups are distinctly different. Beside the age of onset, EOA and AOA groups are also different regarding: motor phenotype, genes involved, genetic mode of inheritance, nature of associated genetic mutations and patterns of disease progression.

Previous studies in patients with AOA have shown that the presence of ataxia with comorbid dystonia (AOAD^+^) concerns a relatively frequently observed clinical phenotype in adulthood-onset ataxias [3,4,5]. Depending on the underlying AOA gene mutation, the percentage of comorbid dystonia (AOAD^+^) may vary between 0% up and 53% [5]. In AOA, the exact pathogenic mechanism for dystonic comorbidity is not fully characterized, yet. Considering the degenerative nature of AOA disease courses, one could assume that extra-cerebellar degeneration may be involved when disorders progress [5]. However, there are also AOA phenotypes that can initially present with dystonia instead of ataxia [6,7,8]. In a previous study, we have explored the converging biological pathways for dystonia and AOA by determining the “shared genetics” between spinocerebellar ataxias (SCA)- and dystonia genes [3]. Forthcoming results indicated that there was a marked over-representation of shared genes involved in GABA-ergic signalling and in neurodevelopment [3]. This implicates that, at least in addition to extra-cerebellar neurodegenerative damage, aberrations in neurotransmission and developmental regulated genes must be involved in the pathogenesis. In line with our previous findings in AOAD^+^, we now aimed to explore the prevalence and underlying pathogenesis in paediatric and young adult patients with EOA, with the underlying hypothesis that EOAD^+^ could be associated with abnormal regulation of developmental genes and aberrations of neurotransmitter pathways as well. In mixed dystonic and ataxic EOAD^+^-phenotypes, we anticipated that pathogenetic insight would contribute to an insightful diagnostic approach.

In the present EOA study, we therefore aimed to elucidate the underlying key biological pathways of dystonic comorbidity (EOAD^+^). We hypothesized that EOAD^+^-phenotypes could be associated with: (1) extra-cerebellar neuro-degenerative alterations determinable by MRI; (2) identifiable shared genetic/molecular pathways determinable by gene co-expression networks in specific EOAD^+^ genotypes; and (3) non-specifically (in-silico) determinable genetic/molecular pathways in shared genes between AOA, EOA and dystonia gene panels, that may induce ataxia and dystonia in a concurrent way. We hypothesized that if comorbid dystonia could be explained by neurodegenerative processes, we would expect an association between the prevalence of comorbid dystonia and disease duration and/or age of the patient, both in our cohort, as well as in literature. This could also implicate a higher prevalence of comorbid dystonia in adult patients with AOA than in young patients with EOA. When the comorbid occurrence of dystonia in EOA would rather be attributable to shared molecular pathways and pathogenetic mechanisms, one would expect potentially corresponding results between two different “genetic-network-analyses” groups: (1) in EOAD^+^ genotypes, with specifically identified dystonic comorbidity; and (2) in non-specifically (in-silico-determined) shared genes in up-to-date with EOA, AOA and dystonia gene panels, that could theoretically cause ataxia and dystonia in a concurrent way [3].

In perspective of the above, we conducted this study in two parts: Part I: in a cohort of 80 EOA-patients, we investigated: (1) the prevalence of EOAD^+^, (2) the association between prevalent EOAD^+^ comorbidity and disease duration and/or age of the patient, and (3) the association between EOAD^+^ comorbidity and patterns of extra-cerebellar MRI abnormalities.

Part II: By genetic network and functional enrichment analysis, we investigated: (1) the shared underlying pathways by determining co-expression networks in the identified EOAD^+^ genotypes; (2) the shared underlying pathways by determining co-expression networks in (in-silico-determined) shared genes between AOA, EOA and dystonia gene lists (panels); and (3) comparative outcomes between specifically identified EOAD^+^ genotypes (from our database) and non-specifically (in-silico determined) shared genes in up-to-date EOA, AOA and dystonia gene lists (panels).

To the best of our knowledge, this is the first study providing a comprehensive approach to explore the prevalence and pathogenesis of EOAD^+^.

## 2. Patients and Methods

The study was carried out following the rules of the Declaration of Helsinki of 1975 (revised in 2013), in accordance with the research and integrity codes of the University Medical Center Groningen (UMCG). The Medical Ethical Committee of UMCG had approved the study (study no. UMCG research register METc 2015/01053, METc approval date 11 July 2012). According to Dutch medical ethical law, both parents and children older than 12 years provided informed consent whereas children younger than 12 years of age provided informed assent for phenotypic assessment.

### 2.1. Phenotypic Assessment of Dystonic Comorbidity in a Cohort of EOA Patients

#### 2.1.1. EOA Database

We included the video-recordings from a cohort of 80 EOA-patients that had visited the paediatric neurology outpatient clinic at UMCG over the last 10 years. Included patients fulfilled the criteria for “EOA”, implicating: symptomatic initiation of ataxia before the 25th year of life or an underlying genetic diagnosis associated with a primary ataxic phenotype, as indicated by the OMIM database (Online Mendelian Inheritance in Man, OMIM. McKusick-Nathans Institute of Genetic Medicine, Johns Hopkins University (Baltimore, MD, USA), 24 December 2016. WorldWide Web: http://omim.org/). In accordance with international criteria for EOA databases, we included patients with congenital, developmental, metabolic, degenerative, and/or unknown causes of ataxia starting before the 25th year of life [9]. Patients were excluded when they exhibited iatrogenic causes, such as underlying infectious, traumatic, intoxicative, cerebrovascular, para- and/or neoplastic pathology [10]. For the underlying diagnosis, age of onset and disease duration of the included patients, see Table 1. The genetic diagnosis of the patients was made using targeted gene panels for either early onset ataxia or dystonia.

#### 2.1.2. Phenotypic Assessment

In accordance with previously described methodology [1], we included videotaped SARA (scale for assessment and rating of ataxia [11]) or ICARS (international cooperative ataxia rating scale [12]) performances, that had been video-taped at the outpatient clinic for patient surveillance reasons. Both scales have been shown to capture paediatric ataxic movement disorder features in a similarly reliable way [13,14]. Furthermore, SARA has been shown to capture other phenotypic features of comorbid movement disorders, as well [15]. We included previously video-taped motor performances of 80 patients fulfilling the criteria of EOA. When patients had been videotaped on several occasions, we systematically included the motor performances that had been performed at the shortest disease duration (i.e., youngest age) of the patient. This provided us the opportunity to assess the motor phenotypes at a relatively early, mostly ambulant disease stage, with the smallest chance of any potential ceiling effects (for instance by the inability to walk or stand). Two paediatric neurologists, specialized in movement disorders, independently phenotyped the videotapes. In accordance with previously described methods, the paediatric neurologists indicated the observed movement disorder features and estimated severity (Appendix A [1]). The assessors individually captured the “print screens” including the time frames from the video-fragments at which they observed dystonic posturing. Patients were assigned to the EOAD^+^ study group when both assessors had indicated that comorbid dystonia was present. Patients were assigned to the EOAD^−^ control group, when both assessors had indicated that comorbid dystonia was absent. In the remaining patients (neither belonging to the EOAD^+^, nor to the EOAD^−^ control group), both assessors explained their phenotypic choice in a separate after-session by play-back at the indicated time frames from the “print screens”.

To allow subsequent statistical comparison on a sufficient number of genes in the study- and control-group, we had to supplement the EOAD^−^ (control) group with additional ataxia genes that were reported without comorbid dystonia, in literature (PubMed and OMIM). For genes included in the EOAD^+^ study and EOAD^−^ control group, see Appendix A.

#### 2.1.3. MRI Abnormalities in EOAD^+^ and EOAD^−^ Subgroups

We subdivided the local cohort of 80 EOA patients into phenotypes with and without comorbid dystonia (i.e., the EOAD^+^ study-group and EOAD^−^ control group, respectively). In both groups, we subsequently associated the underlying genotypes with the corresponding brain abnormalities reported in literature (PubMed and OMIM databases). We characterized cerebral MRI abnormalities in EOAD^+^ and EOAD^−^ groups for: (1) the neuro-anatomical location and (2) the nature of cerebral abnormalities.

### 2.2. Network Analysis

#### 2.2.1. Pathway and Network Analysis on the Study-Group (EOAD^+^) and Control-Group (EOAD^−^)

In the EOAD^+^ study group and EOAD^−^ control group, we related the associated genotypes with the patterns of MRI abnormalities. Subsequently, we performed a pathway and network analysis to evaluate the underlying biological processes and molecular pathways associated with the characterized genetic subgroups. For this purpose, we used the co-expression tool GeneNetwork (www.genenetwork.nl) to generate gene networks using the gene set enrichment feature. The pathway enrichment prediction of the clusters in the disease-specific networks was also performed by GeneNetwork and only the top significant gene ontology (GO) biological pathways were considered. In order to obtain sufficient genes for statistical analysis of the EAOD^−^ control group, we added ataxia genes that were not reported with comorbid dystonia in literature (*ABHD12*; *IFRD1*; *KIAA0226*; *PHYH*; *TDP1*; *VWA3B*; *GTF2H5*; *FLVCR1*; *ACO2*; *HSD17B4*; *DNAJC3* gene mutations; PubMed and OMIM).

#### 2.2.2. Pathway and Network Analysis in EOA, AOA and Dystonia Genes

In order to compare our specific pathway and network results (obtained ad IIa), with the non-specific outcomes derived from (in-silico-determined) shared genes between complete, up-to-date clinically applied gene panels (that could concurrently induce ataxia and dystonia), we compiled the most recent disease associated gene lists (used for clinical genetic diagnostics at the Department of Genetics of the UMCG, Groningen, the Netherlands), including EOA (*n* = 152 genes), AOA (*n* = 80 genes) and dystonia (*n* = 100 genes); see Appendix A. The biological pathways that were enriched in the EOA, AOA and dystonia genes were identified by the Toppfun feature of ToppGene Suite (https://toppgene.cchmc.org). The GO biological pathways were considered significant up to *p*-values 0.005 (Bonferroni, e.g., corrected for multiple testing). In accordance with previously published methods [3], we used GeneNetwork (www.genenetwork.nl) a co-expression tool by integrating 31,499 public RNA-seq samples [16] to generate the EOA, AOA and dystonia gene co-expression networks using the gene set enrichment feature. The pathway enrichment prediction of the clusters in the disease-specific networks was also performed by GeneNetwork and only the top significant GO biological pathways were considered for this work. GO biological pathways were considered significant up to *p*-values of 5 × 10^−5^.

#### 2.2.3. Comparison of Shared Pathways between EOAD^+^ and EOA, AOA and Dystonia Gene Panels (2a versus 2b)

Finally, we compared the underlying shared pathways between: (1) specific EOAD^+^ genotypes that were phenotyped with comorbid dystonia; and (2) (in silico determined) non-specifically shared pathways, derived from up-to-date EOA, and AOA, dystonia gene lists panels, that could concurrently induce ataxia and dystonia. For this purpose, we used the EOA, and AOA, dystonia gene lists that are included in the gene panels at the University Medical Center Groningen.

### 2.3. Statistics

The reliability of the agreement between the observers, was indicated by Cohen’s kappa. Results were interpreted in accordance with Landis and Koch as: poor (*k* < 0); slight (*k* 0–0.20); fair (*k* 0.21–0.40); moderate (*k* 0.41–0.60); substantial (*k* 0.61–0.80) and almost perfect (*k* 0.81–1.00) [17]. We determined normality of disease duration and age of the patient by Shapiro Wilk test. We associated the presence of comorbid dystonia with both disease duration (at the time of the included video-recording) and age of the patient by Mann–Whitney U test. The significance level was set at α = 0.05. Statistical analysis was performed using IBM SPSS statistics 23.0, Statistics for Windows, Version 23.0. Armonk, NY, USA: IBM Corp. In the study and control group, we combined and compared specific groups of genes according to the associated MRI patterns, using the Fisher-exact test. 

## 3. Results

### 3.1. Prevalence of Comorbid Dystonia in 80 Patients with EOA

#### 3.1.1. Clinical Characteristics of Included EOA-Patients

For the underlying diagnosis, age of onset, disease duration of the included patients, see Table 1. The disease duration and age of the patient (at video-assessment) were not normally distributed (Shapiro Wilk test (*p* = 0.001)). In 84% (67/80) EOA patients, the underlying association with the disease symptom ataxia was confirmed by genetic, metabolic and/or radiologic findings. In 78/80 (98%) of the recorded EOA-patients, either one of the two observers had recognized the presence of the symptom ataxia. In 76/80 (95%) of the recorded EOA-patients, both observers had recognized the presence of the symptom ataxia. The two patients in whom none of the observers had recognized ataxia, were diagnosed with an *ATP1A3* and *TUBB2A* mutation, respectively. Both patients had been described with ataxic features in the records of the outpatient clinic, but these features could apparently not be identified during the off-line video-assessment of the specific SARA video-recording. For rough scoring data and specific gene mutations, see Appendix A. Scored dystonic comorbidity is indicated in Appendix A.

#### 3.1.2. Evaluation of Comorbid Dystonia

In 52/80 (65%) of the EOA-patients, comorbid presence of dystonia was indicated by both observers, characterized by “comorbid dystonia”. In 11/80 (14%) of the EOA-patients, the symptom dystonia was assessed by one observer and in 17/80 (21%) dystonia was assessed as absent by both observers. In 3/52 (6%) of the EOA-patients with comorbid dystonia (*TTPA*, *ATP1A3* and *TUBB2A* gene mutations), both observers had indicated that dystonia was severely present and that dystonia was presented as the main phenotype. In two of these patients (*ATP1A3* and *TUBB2A* gene mutations), ataxia had not been identified. In the other 49/52 (94%) of EOA-patients with comorbid dystonia, both observers had indicated that dystonia was (mostly mildly) present and that dystonia concerned the secondary phenotype. Either presence, or absence of comorbid dystonia was not significantly associated with EOA disease duration and/or age of the patient (*p* = 0.645 and *p* = 0.103, respectively; Mann–Whitney U test), see Appendix A. In the patients with successive video-recordings, dystonic features did not longitudinally change from mild to severe (data not shown).

#### 3.1.3. Association between Phenotype and Underlying Etiology

EOAD+ phenotypes were associated with genetic mutations (*n* = 41; 79%), congenital malformations of the fossa posterior (*n* = 2; 4%) and unknown causes (*n* = 9; 17%), see Table 2a. The EOA phenotypes without comorbid dystonia (28/80; 35%) were associated with genetic mutations (13; 81%); congenital malformations of the fossa posterior (*n* = 1; 6%), and unknown causes (*n* = 2; 13%), see Table 2b. The diagnoses Friedreich’s ataxia, North Sea progressive myoclonus epilepsy, episodic ataxia type 2 and congenital malformations of the fossa posterior were both associated with presence and with absence of comorbid dystonia (EOAD^+^ and EOAD^−^ phenotypes); see Table 2a,b, respectively.

#### 3.1.4. Reliability of Agreement between the Observers

The reliability of the agreement between the observers, was indicated by Cohen’s kappa of 0.668 (*p* < 0.001). The kappa value was interpreted as sufficient to good in accordance with Landis and Koch [17]. In 69/80 (86%) patients, there was full agreement between the two observers on the presence or absence of comorbid dystonia. In 11/80 (14%) patients, the presence of comorbid dystonia was only indicated by one observer. In these 11 patients, the other observer had explained that the dystonic-like features were recognized, but that these features could not be discriminated from dystonic-like features due to physiologic immaturity of the central nervous system. These cases were therefore excluded from the subsequent analysis of EOAD^+^ and the EOAD^−^ groups.

#### 3.1.5. EOAD^+^ and EOAD^−^ Groups and Associated MRI Abnormalities

In the investigated cohort, there were 25 genotypes in association with EOAD^+^, 6 genotypes in association with EOAD^−^ and 4 genotypes in association with both EOAD^+^ and EOAD^−^. In total, only 2 genotypes were included in the EOAD-control group. We, therefore, supplemented the EOAD- control group with ataxia genotypes that were not reported with comorbid dystonia in literature (including *ABHD12*; *IFRD1*; *KIAA0226*; *PHYH*; *TDP1*; *VWA3B*; *GTF2H5*; *FLVCR1*; *ACO2*; *HSD17B4*; *DNAJC3* gene mutations; PubMed and OMIM). For the included EOAD^+^ and EOAD^−^ groups and associated MRI abnormalities, see Appendix A. Reported MRI abnormalities were subdivided into hypoplasia; atrophy; and specifically described damage (see also Appendix A). Associating reported MRI abnormalities with EOAD^+^ and EOAD^−^ phenotypes, revealed a significant association between EOAD^+^ phenotypes and abnormalities at the pons and/or basal ganglia and/or thalamus (*p* = 0.001), see Appendix A. Comparing the division of white and grey matter damage between EOAD^+^ versus EOAD^−^ groups, did not reveal statistical differences, see Appendix A.

### 3.2. Pathway and Network Analysis

#### 3.2.1. EOAD^+^ Genotypes

In the EOAD^+^ gene group, pathway analysis revealed the strongest enrichment for GO biological processes involved in organelle organization (*p* = 8.853 × 10^−17^), and additionally in cellular component organization or biogenesis (*p* = 2.315 × 10^−12^), chromosome organization (*p* = 7.158 × 10^−8^) and cytoskeleton organization (*p* = 3.441 × 10^−7^). These are cellular processes resulting in the assembly, (re-) arrangement or disassembly of organelles, cellular components, chromosomes and cytoskeleton in a cell. For pathway and network analysis in EOAD^+^ and EOAD^−^ groups in association with allocated MRI damage, see Table 3.

#### 3.2.2. Shared Genes in EOA, AOA and Dystonia Gene-Lists (Panels)

In EAO, AOA and dystonia gene lists (Appendix A), we identified 54 shared genes between EAO and AOA, 13 between EAO and dystonia, and 8 between AOA and dystonia (Appendix A and Appendix A). The latter 8 genes were also shared between EAO and dystonia (i.e., shared between EOA, AOA and dystonia). These gene mutations included: *ATP1A3* (associated with the expanding phenotypic spectrum of alternating hemiplegia of childhood, rapid-onset dystonia-parkinsonism, *CAPOS* and *FIPWE*) [18], *POLG* (mitochondrial depletion syndrome), *NPC1* (Niemann–Pick disease, type C1), *TUBB4A* (DYT4), *MTTP* (abetalipoproteinemia), *SPG7* –(spastic paraplegia 7) and *SLC2A1* (GLUT1 deficiency syndrome). Two of these genes, *NPC1* and *MTTP*, are associated with plasma lipoprotein particle organization and cholesterol homeostasis.

#### 3.2.3. Pathway Analysis in EOA, AOA and Dystonia Gene Lists (Panels)

We identified 90 significant GO biological pathways in EOA, 39 in AOA and 132 in dystonia genes (Appendix A–S10). Of these pathways, 8 were shared between the three disorders, including cation- and ion transport, cation- and ion transmembrane transport, inorganic cation- and ion transmembrane transport, transmembrane transport, and locomotion pointing to an important role of cellular communication via synaptic transmission and movement in the underlying shared biology. For EOA, the most enriched GO pathways were locomotion, neurogenesis, myelination, and ion transport. For AOA, similar to EOA, pathways were associated with locomotion, ion- and trans-membrane transport, chemical- and synaptic transmission, and anterograde trans-synaptic signaling. For dystonia, pathways such as cellular respiration, oxidation-reduction process, respiratory- and electron transport chain, mitochondrial respiratory chain complex assembly and mitochondrion organization were identified. Overall, EOA and AOA are more similar in their underlying biological pathways compared to either one of them with dystonia, whereas dystonia shared only a few (*n* = 5) unique biological pathways with EOA but not with AOA.

#### 3.2.4. Network Analysis in EOA, AOA, Dystonia

The EOA, AOA, dystonia networks comprised of 7, 3, and 4 clusters, respectively (Appendix A). We identified 472 shared genes between the three networks (Figure 1A and Supp. Appendix A). The networks of EOA-AOA showed most overlap in genes (*n* = 1210), compared to EOA-dystonia (*n* = 1004) and AOA-dystonia (*n* = 500). The 472 shared genes between the three networks were enriched for GO pathways (top 10 ToppGene) involved in carboxylic acid—and organic acid metabolic process, fatty acid—and lipid metabolic process, and organic—and carboxylic acid catabolic process (Figure 1B). The 532 uniquely shared genes between EOA and dystonia were enriched for GO pathways involved in cellular respiration, drug metabolic process, ATP biosynthetic process, respiratory electron chain transport and purine ribonucleoside triphosphate biosynthetic process (Appendix A), whereas the 28 uniquely shared genes between AOA and dystonia were enriched in GO pathways involved in the release of calcium into the cytosol and calcium ion transport (Appendix A).

## 4. Discussion

To the best of our knowledge, this is the first study targeting at the underlying biological pathways in patients with EOA with comorbid dystonia (EOAD^+^-phenotypes). In the majority of EOA-patients, we observed only mildly dystonic features. The prevalence of dystonic comorbidity (65%) was apparently higher than previously reported prevalence in AOA-patients (0% to 53%, depending on the genotype) [5]. In addition to MRI abnormalities at the cerebellum, EOAD^+^-phenotypes revealed a strong association with MRI abnormalities at the basal ganglia and/or thalamus and/or pons (implicating disturbed signaling somewhere in the anatomical cortico-basal-ganglia-ponto-cerebellar network [19,20]). There was no association between the presence or absence of comorbid dystonia and EOA disease duration and/or age of the patient, implicating that other factors than ongoing neuro-degeneration are likely to play a role in the pathogenesis of comorbid dystonia. In our EOAD^+^-study group, pathway and molecular co-expression network analysis indicated an underlying association with organelle and cellular organization (underlying energy production and signal transduction). As such pathways are not implicated in EOA alone, these findings are in line with previous studies demonstrating a pathophysiologic role for cytoskeletal reorganization in the underlying biology of dystonia [21]. Comparing these results with (in-silico-determined) network analysis in shared EOA, AOA and dystonia gene lists (panels), showed enrichment for Krebs-cycle (tricarboxylic acid cycle (TCA)) and fatty acid/lipid metabolic process, underlying the concept of hampered energy production and signal transduction. From these data, we conclude that both specifically (EOAD^+^) and non-specifically (in silico determined) shared pathways and networks analyses implicate an underlying role for cellular energy production and network signal transduction in the pathogenesis of EOA with comorbid dystonia. This may have implications for genetic testing. Instead of testing with a single EOA gene panel, one may consider using Whole Exome Sequencing (WES), a complete movement disorder panel and copy number variation analysis, whereas diagnostics by Whole Genome Sequencing (WGS) may have a wider application, in the future. Previous studies in neuro-degenerative AOA disorders, have implicated that the comorbid presence of dystonia should be regarded as an expression of the same disease continuum [4,22,23]. Conversely, in adult patients, dystonic symptoms have also been associated with cerebellar pathology [24,25] and cerebellar symptoms, including action induced tremors [26], eye blink conditioning [27] and saccadic adaption [28]. However, in the presently studied cohort of 80 relatively young EOA patients, we observed comorbid dystonia (EOAD^+^) in the majority (65%) of patients. This EOAD^+^ subgroup revealed a large heterogeneity in genotype-phenotype relationships, reflected by: (1) identical genetic mutations that were associated with EOAD^+^ and also with EOAD^−^ phenotypes (in different patients), (2) absence of EOAD^+^ features in genotypes that have been identified with comorbid dystonia in literature and (3) presence of EOAD^+^ features in EOA genotypes that have not been reported with comorbid dystonic features, before. As expected, we observed that MRI abnormalities of the basal ganglia and/or pons and/or thalamus were associated with the EOAD^+^ phenotype. In addition to the well-known association between abnormalities at the basal ganglia and thalamus with dystonia, the pedunculo-pontine tegmental nucleus (PPTg) at the pons has been shown to connect between the basal ganglia and cerebellar nuclei and thalamus [29] implicating that hampered signaling in the anatomical cortico-basal-ganglia-ponto-cerebellar network may be involved [19,20].

Comparing the dystonic prevalence in “early disease onset” EOA (EOAD^+^; 65%) with previously reported “adult disease onset” AOA (AOAD^+^; 0% to 53% depending on the genotype) [5,30,31,32,33], reveals a higher prevalence in the first group. This could be theoretically attributed to several factors. First, it is well known that dystonic-like features may physiologically appear in young children, due to the incomplete maturation of the central nervous system [34,35]. However, in EOA we observed no association between EOAD^+^ and young age and/or shorter disease duration. Furthermore, the majority of patients were older than 10 years of age. After this age, physiologic dystonic-like features have mostly disappeared [34,35] and, finally, we had excluded all patients with doubtful minor developmental dystonic-like features from the study. Second, one could attribute the higher prevalence of dystonic comorbidity in EOA than in AOA to more advanced extra-cerebellar neuro-degeneration. However, considering the younger age of the EOA patients and the inclusion of the firstly recorded movement disorder performances, this appears unlikely, as well. Another, and much more likely explanation is provided by our non-specific, in silico pathway and network analysis, performed on shared genes between up-to-date EOA, AOA and dystonia gene panels. Comparing gene network similarities, revealed about twice as much overlapping gene networks between EAO- and dystonia-genes than between AOA- and dystonia-genes. From this molecular genetic perspective, it could be derived that dystonic comorbidity is also about twice as likely to concur with EOA than with AOA. 

In the present study, we hypothesized that the underlying genetic mechanisms for EOAD^+^ could both involve: (1) shared pathways inducing a specific EOAD^+^ phenotype, and/or (2) non-specifically shared pathways by genes that may be concurrently expressed in EOA, AOA and dystonia, inducing comorbid features. Investigating shared pathways in the specific EOAD^+^ group, revealed an association with organelle- and cellular- organization. Until now, these pathways have not been described in EOAD^+^ before. Mitochondria are important organelles generating most of the cellular energy by the TCA (Krebs cycle). Pathways of cellular organizations are involved in the axonal cytoskeleton providing the basis for axonal transport and network signaling. This may imply that novel EOAD^+^-phenotype related gene mutations could be found in association with these molecular pathways.

By investigating the non-specifically shared genes and pathways between EOA and AOA genes, we identified quite similar pathway enrichment for the EAO and AOA gene list, that was different from dystonia. The top biological pathways observed for EAO and AOA were involved in locomotion and neurogenesis. These biological pathways have also been implicated in the pathogenesis of ataxia syndromes [36,37]. The top biological pathways observed for dystonia were involved in cellular respiration and metabolism. Additionally, studies reported changes in cellular—and /or mitochondrial respiration in dystonia [38,39], supporting the validity of our in silico genetic analysis. Not surprisingly, none of the top pathways underlying either ataxia or dystonia were shared between EAO, AOA and dystonia. In fact, several pathways involved in cation and ion membrane transport were enriched in the common genes, pointing to an important role for neuronal communication that is also consistent with prior knowledge on the pathology of these mixed disorders [38,40,41]. Furthermore, we observed that carboxylic acid—and organic acid metabolic and catalytic processes were enriched in the common genes, pointing out to the tricarboxylic acid cycle (TCA), or Krebs cycle. The TCA cycle, is essential for mitochondrial ATP production and is fueled by fatty-acid–oxidation. Furthermore, the TCA cycle is crucial for the synthesis of gamma aminobutyric acid (GABA), the main neurotransmitter of Purkinje cells (PCs). In dystonic syndromes, it is reported that PCs are dysfunctional and in ataxic syndromes PCs are often also degenerative [25,42]. Of note, one of the clusters of the AOA network was enriched for genes involved in gamma-aminobutyric acid (GABA) signaling pathway, and altered GABA-ergic signaling has also been reported to play a role in patients with cervical dystonia [43]. Finally, cellular energy failure has been implicated in the pathogenesis of cerebral demyelination. Whether preferential loss of Myelin-associated glycoprotein is a feature of primary mitochondrial disorders [44], or due to mutations in nuclear genes is still unclear [45].

The enrichment for lipid and fatty acid homeostasis in the shared genes (Figure 1B) of the disease specific molecular networks further support the role for development of the central nervous system in the pathology of these disorders. Cholesterol is an essential lipid for mammalian cells, and is necessary for the numerous formations of efficient synapses, which stems from de novo synthesis [46]. Whereas fatty acids and their metabolites are required for normal brain development and the activation of gene transcription regulating long-chain polyunsaturated fatty acids formation. Many neurodegenerative diseases are associated with disrupted lipid- and cholesterol homeostasis, including such Niemann Pick type C disease, Smith Lemli Opitz, and SCA3 [46,47]. In neurodegenerative mouse models for spinocerebellar ataxia, it was shown that impaired cholesterol metabolism reduces the Purkinje cell number and induces motor coordination deficits [48]. Furthermore, it has been shown that that the cerebellum can modulate the basal ganglia activity [19,20] by input from the neurologic cerebello-thalamo-basal ganglia anatomical pathway [49]. In the central nervous system, oligodendrocytes generate multiple layers of myelin around axons of the central nervous system to enable fast and efficient nerve conduction. Until recently, saltatory nerve conduction was considered the only purpose of myelin, but myelinating oligodendrocytes can also provide metabolic support to neurons, and regulate ion and water homeostasis by adapting to activity-dependent neuronal signals [50]. Mutations in very long chain fatty acid elongase 4 and 5 (Elovl4 and ElovL5) are reported to cause spinocerebellar ataxia [51,52,53] and accumulation of the branched-chain acid fatty acid was reported to be associated with Refsum disease caused by mutations in Phytanic acid alpha-oxidation (in AOA gene panel) [54]. Additionally, MECR mutations cause a mitochondrial fatty-acid synthesis disorder and is characterized by a childhood-onset dystonia [55]. Altogether, these crucial biological pathways may thus support the hypothesis that they can concurrently underlie the initiation of ataxia and dystonia [4]. Nevertheless, this does not necessarily implicate that these pathways also play a specifically causative role in the pathogenesis of comorbid dystonia. However, investigating the pathway and network analysis in the specifically phenotyped EOAD^+^ group, reveals a similar role for organelle and cellular organization in dystonic comorbidity. Although not identical, both specifically and non-specifically shared pathways may thus implicate an association with hampered cellular energy production and network signal transduction.

We are aware of some weaknesses to this study. In the first place, the presently studied EOA gene panel cannot be considered complete, since new genes are being, and will be added in the future. Furthermore, by using an EOA database from a single center, we cannot exclude local influences on the outcome data. For instance, we noticed that some of the EOAD^+^ genes are not associated with dystonia in literature, and vice versa. However, considering the fact that (1) EOA is a rare disorder, (2) we were able to include a considerable cohort of 80 EOA patients, (3) the identified EOAD^+^ phenotypes were linked with extra-cerebellar MRI alterations at the basal-ganglia-ponto-thalamic network, and (4) pathway- and network-analyses in both specific EOAD+ phenotypes and in silico determined shared genes reflected similarly underlying biological processes, we would suggest that the present results can be interpreted as indicative. Hopefully, future collaboration with European and even world-wide based ataxia databases will elucidate this.

In summary, in a local cohort of 80 EOA-patients, we observed dystonic comorbidity in the majority of patients. Exploration of the underlying clinical, anatomical and biological pathways revealed shared pathophysiology, despite genotype-phenotype heterogeneity. Both patient specific (in EOAD^+^) and non-specific (in silico determined) pathway- and network-analyses implicated associated biological pathways involved in organelle and cellular organization, respectively in TCA cycle processes and lipid and fatty-acid homeostasis. Both outcomes suggest that hampered energy production and network signal transduction may play an underlying role in the pathophysiology of ataxia with comorbid dystonia.

These findings may have important implications for the diagnostic approach in mixed “EOA” comorbid dystonia movement disorders. Since network analyses in both specifically determined EOAD^+^-genotypes and also in non-specifically, in silico, determined shared genes in EOA, AOA and dystonia panels both refer to similar underlying pathways, one may hypothesize the presence of a common pathogenesis. This would implicate that EOAD^+^-phenotypes can be concurrently induced by shared genetic networks between EOA and dystonia genes. This could explain the heterogeneous genotype-phenotype relationships varying from predominant ataxia at one end of the spectrum, continuing with ataxia and comorbid dystonia, and, finally predominant dystonia at the other end of the spectrum.

Altogether, in perspective of: (1) the high prevalence of EOAD^+^ phenotypes, (2) the heterogeneity of genotype-phenotype relationships, (3) the shared anatomical pathways and (4) the shared underlying biological pathways that contribute to the same disease continuum, it might be a rationalistic approach to test EOA patients with a complete, up to date movement disorder panel (including EOA and dystonia gene lists), instead of with a single EOA gene panel. In the future, we aim to investigate the pathogenesis of other mixed EOA phenotypes by determining shared pathways between EOA and other comorbid movement disorders, as well.

## 5. Conclusions

Comorbid dystonia is prevalent in the majority of EOA patients. The underlying biological pathways can be linked with energy depletion and hampered signal transduction involving the cortical-basal-ganglia-pontine-cerebellar network. Hopefully, future insight in the underlying processes causing the heterogeneous, mixed EOA phenotypes may contribute to the yield of diagnostic testing and innovative therapeutic strategies.

## Figures and Tables

**Figure 1 diagnostics-10-00997-f001:**
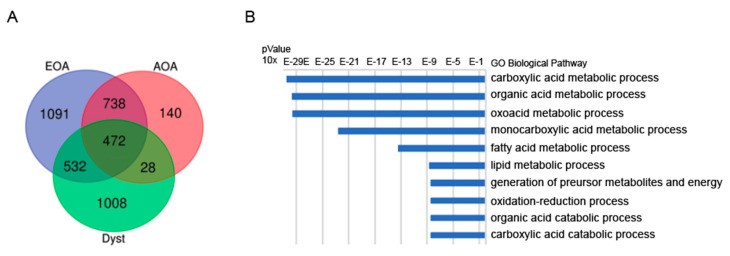
Shared genes and pathways between EAO, AOA and dystonia networks. Legend: (**A**) Venn diagram plot showing 472 common genes between EOA, AOA and dystonia. The gene networks of EOA—dystonia (*n* = 1004 (532 + 472)) reveal more overlap than the gene networks between AOA—dystonia (*n* = 500 (28 + 472)), suggesting that the gene networks of EOA is more similar to the dystonia network compared to the network of AOA—dystonia. (**B**) Top 10 of the most enriched pathways (top 10 ToppGene) of the common genes between EOA, AOA and dystonia. The most enriched pathways involved are involved in carboxylic acid—and organic acid metabolic process and fatty acid—and lipid metabolic process.

**Table 1 diagnostics-10-00997-t001:** Early onset ataxia (EOA) patient information.

Case	Age of Onset (Year)	Duration (Full Years)	Age at Assessment (Year)	Gene Name *	Mutation Type	Neurological Diagnosis
1	0	14	14	*RELN*	VUS	cerebel cort dyspl, hypopl pons
2	0	17	17	*LAMA1A*	MM	Poretti Boltzhausen syndrome
3	0	13	13	-	-	Dandy Walker malformation
4	0	14	14	Unknown	-	Unknown
5	0	9	9	*SOX 10*	MM	Shah-Waardenburg syndrome
6	0	22	22	*CHD7*	MM	CHARGE Syndrome
7	7	6	14	Unknown	-	Unknown
8	0	7	7	*KIAA0586*	MM	Joubert Syndrome 23
9	11	0	11	-	-	Cediak Higashi
10	0	12	12	*SPTBN2*	Del, MM	SCA5
11	3	7	11	Unknown	-	Unknown
12	2	8	10	*FXN*	GAArepeat	Friedreich’s ataxia
13	0	10	10	*CTNNB1-gen*	MM	AD MR 19
14	0	8	9	*KCNC3*	MM	SCA13
15	0	9	10	Unknown	-	Unknown
16	1	7	8	*HSD17B10*	MM	MHBD-deficiency
17	4	5	9	*FXN*	GAA repeat	Friedreich’s ataxia
18	3	4	7	*EBF3 mutation*	MM	HADDS syndrome
19	4	1	5	*FXN*	GAA repeat	Friedreich’s ataxia
20	0	0	1	*INPPE5*	MM	Joubert syndrome type 1
21	14	1	16	Unknown	-	Unknown
22	5	6	11	*GOSR2*	MM	Northsea progr myocl
23	2	4	6	Unknown	-	Unknown
24	0	5	5	Unknown	-	Unknown
25	2	8	10	Unknown	-	Unknown
26	13	2	15	*CACNA1A*	MM	Episodic Ataxia type 2
27	4	7	11	*FXN*	GAA repeat	Friedreich’s ataxia
28	2	5	7	*KCND3*	MM	SCA19
29	6	8	14	*CACNA1A*	MM	Episodic Ataxia type 2
30	1	2	3	*CAMTA1*	MM	CAMTA1
31	0	13	13	Unknown	-	Unknown
32	2	7	9	*TITF1*	MM	Benign Hereditary Chorea
33	4	2	6	*ZMYND11*	MM	AD, MR type 30
34	1	12	13	*ITPR1*	MM	SCA 29
35	3	9	12	*ITPR1*	MM	SCA29
36	4	11	15	*ITPR1*	MM	SCA29
37	12	0	12	Unknown	-	Unknown
38	1	1	2	Unknown	-	Unknown
39	6	3	9	*SPTBN2*	Del, MM	SCA5
40	1	7	8	*ATP1A3*	MM	RDP-AHC-Atax
41	2	5	8	*ATP1A3*	MM	AHC
42	9	23	32	*TTPA*	MM	AVED
43	4	11	15	*FXN*	GAA repeat	Friedreich’s Ataxia
44	12	3	15	*NPC*	MM	Niemann Pick
45	12	22	34	*TTPA*	MM	AVED
46	1	25	26	*T8993G*	MM	NARP
47	16	11	28	*TTPA*	MM	AVED
48	5	11	16	*HTT*	CAG repeat	Juvenile Huntington
49	1	18	19	*ATM*	MM	Ataxia Telangiectasia
50	5	3	8	*FXN*	GAA repeat	Friedreich’s Ataxia
51	0	5	5	-	-	cong malf fossa pos
52	11	6	18	*mtDNA*	MM	Kearns Sayre Syndrome
53	10	2	13	*FXN*	GAA repeat	Friedreich’s ataxia
54	14	3	18	*SPG-11*	MM	HSP
55	1	17	18	*GOSR2*	MM	Northsea progr myocl
56	2	23	25	*GOSR2*	MM	Northsea progr myocl
57	0	6	6	Unknown	-	Unknown
58	3	13	16	*CACNA1A*	CAG repeat	Episodic Ataxia type 1
59	0	15	15	*KIAA0586*	MM	Joubert Syndrome 23
60	3	3	6	*GOSR2*	MM	Northsea progr myocl
61	3	13	16	*CACNA1A*	CAG repeat	Episodic Ataxia type 1
62	13	9	22	*TTPA*	MM	AVED
63	2	0	3	*GOSR2*	MM	Northsea progr myocl
64	2	18	20	*GOSR2*	MM	Northsea progr myocl
65	2	1	3	*ALDH3A2*	MM	SjogrenLarsson
66	8	5	13	*SPG11*	MM	Spastic paraplegia 11
67	6	13	19	*FXN*	GAA repeat	Friedreich’s Ataxia
68	7	14	21	*FXN*	GAA repeat	Friedreich’s Ataxia
69	9	13	22	*FXN*	GAA repeat	Friedreich’s Ataxia
70	6	10	17	*FXN*	GAA repeat	Friedreich’s Ataxia
71	4	10	14	*FXN*	GAA repeat	Friedreich’s Ataxia
72	5	7	12	*FXN*	GAA repeat	Friedreich’s Ataxia
73	2	9	11	*TUBB2A*	MM	CDCBM5
74	1	1	3	*ATP1A3*	MM	FIPWE
75	1	11	12	*CACNA1A*	CAG repeat	Episodic Ataxia type 2
76	7	5	12	*ATXN7*	CAG repeat	SCA7
77	15	0	16	*SLC2A1 gen*	MM	Glut-1 def
78	0	3	3	Unknown	-	Unknown
79	5	4	9	*FXN*	GAA repeat	Friedreich’s ataxia
80	6	10	16	*FXN*	GAA repeat	Friedreich’s Ataxia

Gene name * = gene name, mutations are specified in the Appendix A; cerebel cort dyspl = cerebellar cortical hypoplasia; hypopl = hypolasia; VUS = variant of unknown significance; MM = missense mutation; MHBD = 2-methyl-3-hydroxybutyryl-CoA-hydrogenase deficiency, HADDS = hypotonie; ataxie and delayed development syndrome; RDP-AHC-Atax = disease continuum of rapid onset parkinsonism (RDP); alternating hemiplegia of childhood (AHC); ataxia AVED = Ataxia with isolated vitamin E deficiency; NARP = neuropathy; ataxia and retinitis pigmentosa; cong malf fossa pos = congenital malformation fossa posterior; CDCBM5 = cortical dysplasia, complex, with other brain malformations; FIPWE = fever-induced paroxysmal weakness and encephalopathy.

**Table 2 diagnostics-10-00997-t002:** EOA gene mutations with (**a**) and respectively without comorbid dystonia (**b**).

**a.** with comorbid dystonia			
Gene mutation			
*TUBB2A* (*n* = 1)	*ATXN7* (*n* = 1)		*LAMA1A* (*n* = 1)
*FTX* (*n* = 9)	*KCNC3* (*n* = 1)		*CHD7* (*n* = 1)
*INPPE5* (*n* = 1)	*ATM* (*n* = 1)		*LYST* (*n* = 1)
*ATP1A3* (*n* = 3)	*CAMTA1* (*n* = 1)		*HSD17B10* (*n* = 1)
*TTPA* (*n* = 3)	*NARP* (*n* = 1)		*HADDS* (*n* = 1)
*CACNA1A* (*n* = 3)	*ZMYND11* (*n* = 1)		*CTNNB1* (*n* = 1)
*GOSR2* (*n* = 2)	*ALDH3A2* (*n* = 1)		*HTT* (*n* = 1)
*SPTBN2* (*n* = 2)	*TITF1* (*n* = 1)		*SPG11* (*n* = 1)
*KIAA0586* (*n* = 2)	*NPC* (*n* = 1)		* unknown (*n* = 12)
	**b.** without comorbid dystonia		
	Gene Mutation		
*KCND3* (*n* = 1)		*CACNA1A* (*n* = 2)	
*FTX* (*n* = 3)		*SPG11* (*n* = 1)	
*GOSR2* (*n* = 3)		** unknown (*n* = 3)	
*ITPR1* (*n* = 3)			

Legends: * unknown (*n* = 12) = unknown/absent gene mutation in association with malformation of fossa posterior (*n* = 2); *LYST* = Cediak Higashi syndrome (*n* = 1); no clinical diagnosis (*n* = 9); ** unknown (n=3) = unknown/absent gene mutation in association with malformation of fossa posterior (*n* = 1); no clinical diagnosis (*n* = 2). The gene mutations *CACNA1A*, *FTX* and *GOSR2* were present in clinical cases with and without comorbid dystonia. Cases with a congenital malformation of the fossa posterior were both associated with and without comorbid dystonia.

**Table 3 diagnostics-10-00997-t003:** Top biological pathways in Early Onset Ataxia and Dystonia (EOAD^+^) and (EOAD^−^).

Subgroup	Most Significant Pathways	*p*-Value
EOA, Dystonia + (EOAD^+^)	1. organelle organization	8.853 × 10^−17^
	2. cellular component organization or biogenesis	2.315 × 10^−12^
	3. cellular component organization	1.767 × 10^−11^
	4. chromosome organization	7.158 × 10^−8^
	5. cytoskeleton organization	3.441 × 10^−7^
EOA, Dystonia − (EOAD^−^)	1. small molecule metabolic process	1.091 × 10^−17^
	2. cellular lipid metabolic process	2.773 × 10^−15^
	3. lipid metabolic process	2.866 × 10^−15^
	4. cellular lipid catabolic process	6.840 × 10^−15^
	5. carboxylic acid metabolic process	1.376 × 10^−14^
EOA, White Matter damage + (EOAW^+^)	1. organelle organization	4.102 × 10^−8^
	2. cellular component organization	6.759 × 10^−8^
	3. cellular component organization or biogenesis	8.330 × 10^−8^
	4. regulation of organelle organization	4.108 × 10^−7^
	5. regulation of cellular component organization	6.354 × 10^−6^
EOA, White matter damage − (EOAW^−^)	1. ribonucleoprotein complex biogenesis	1.897 × 10^−6^
	2. cellular nitrogen compound metabolic process	3.662 × 10^−6^
	3. ribosome biogenesis	8.758 × 10^−6^
	4. cellular component organization or biogenesis	9.220 × 10^−5^ *
	5. RNA processing	1.389 × 10^−4^ *
EOA, extracerebellar damage + (EOAX^+^)	1. carboxylic acid metabolic process	5.703 × 10^−10^
	2. oxoacid metabolic process	9.228 × 10^−9^
	3. organic acid metabolic process	1.635 × 10^−8^
	4. cellular lipid catabolic process	1.742 × 10^−6^
	5. vacuolar transport	3.896 × 10^−6^
EOA, extracerebellar damage − (EOAX^−^)	1. RNA metabolic process	4.933 × 10^−16^
	2. mRNA metabolic process	1.716 × 10^−14^
	3. nucleic acid metabolic process	2.453 × 10^−13^
	4. gene expression	9.775 × 10^−13^
	5. mRNA processing	5.158 × 10^−12^
EOA, cerebellar damage + (EOAC^+^)	1. cellular component organization	9.435 × 10^−11^
	2. cellular component organization or biogenesis	1.286 × 10^−10^
	3. organelle organization	9.573 × 10^−10^
	4. cellular localization	3.517 × 10^−7^
	5. vacuolar transport	3.177 × 10^−6^
EOA, cerebellar damage − (EOAC^−^)	1. No statistical significant pathways could be found.	
EOA, dystonia+, White matter damage + (EOAD^+^W^+^)	1. organelle organization	9.603 × 10^−15^
	2. cellular component organization or biogenesis	3.714 × 10^−13^
	3. cellular component organization	1.062 × 10^−12^
	4. cellular localization	2.485 × 10^−8^
	5. microtubule-based process	1.631 × 10^−7^

+ = comorbid sign is present; − = comorbid sign is absent; EOAD = EOA and comorbid dystonia; EOAW = EOA and white matter damage; EOAX = EOA and extra-cerebellar damage; EOAC = EOA and cerebellar damage; EOADW = EOA, dystonia and white matter damage. * Not significant. Statistical significance for pathway analysis: *p* < 5 × 10^−5^.

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
