# Peer review of "Early Onset Ataxia with Comorbid Dystonia: Clinical, Anatomical and Biological Pathway Analysis Expose Shared Pathophysiology"

_diagnostics, 2020, doi:10.3390/diagnostics10120997_

Round 1

Reviewer 1 Report

This is an original and elegant study that provides once more evidence for the intimate relationship between ataxia and dystonia, also attempting to clarify the basis of such association.

There are no major concerns.

Only a couple of comments.

The cohort does not include patients with COQ8A variants, a condition accounting for EOA+dystonia (DOI: 10.1016/j.parkreldis.2019.09.015; DOI: 10.1002/ana.25751), which may suggest for further molecular pathways underlying the ataxia-dystonia association.

Authors correctly indicated the role of cerebellum-basal ganglia-cortical network impairment as the pathophysiological substrate of ataxia-dystonia association. Supportive data could be found in doi: 10.1038/nn.2753 and doi: 10.1097/WCO.0000000000000580.

Author Response

The cohort does not include patients with COQ8A variants, a condition accounting for EOA+dystonia (DOI: 10.1016/j.parkreldis.2019.09.015; DOI: 10.1002/ana.25751), which may suggest for further molecular pathways underlying the ataxia-dystonia association.

Answer: We thank the reviewer for this remark and suggestion. The reviewer is correct. We are aware that our gene panel for early onset ataxia may be incomplete as other gene mutations are now being included, and new mutations will be included in the future, as well. However, for COQ8A gene mutations, we would not expect that inclusion in the EOA gene panel analysis would have changed our results, since COQ8A gene mutations are associated with mitochondrial/metabolic pathway mechanisms, indicative of energy failure (which is an underlying pathway observed in the presented EOAD+ group). Furthermore, COQ8A‐ mutation related brain damage is not limited to the cerebellum as well, since it includes the brainstem, supra-tentorial regions and the pons. In this perspective, we would speculate that inclusion of COQ8A‐ gene mutations would fit in with the described outcomes in the EOAD+ group.
- In accordance with the reviewer, we have now added a statement that we are aware that our EOA gene panel cannot be considered as complete, as new gene mutations are likely to be added in the future. See discussion lines 491-492.

Authors correctly indicated the role of cerebellum-basal ganglia-cortical network impairment as the pathophysiological substrate of ataxia-dystonia association. Supportive data could be found in doi: 10.1038/nn.2753 and doi: 10.1097/WCO.0000000000000580.

Answer: We thank the reviewer for this remark and suggestion. We have added these references to the discussion (reference 19 and 20).

Reviewer 2 Report

Dear Authors,

I have read with very interest your interesting manuscript titled “Early Onset Ataxia with Comorbid Dystonia: Clinical, Anatomical and Biological Pathway Analysis Expose Shared Pathophysiology”. The manuscript from Sival and colleagues investigated the prevalence and pathogenesis of EOAD+ in 80 EOA-patients. The main novelty of the study is the EOAD+ cohort. The study is well conducted, and well written. Few suggestions for the authors to improve their manuscript.   

1- the power of study is lack; please could you evaluate if your data could be generalized also at other populations?

2- The authors conclude Line 513-514 that “it might be a rationalistic approach to test EOA patients with a complete, up to date movement disorder panel (including EOA and dystonia gene lists), instead of with a single EOA gene panel”. That is a good point and the discussion could be opened on the need of WGS on this group of patients followed by a complete up-to-date movement disorder gene list analysis. Moreover, Copy number variation analysis would also be interesting in EAOD+ cohort.

3- Table1: It would be useful to include more information about the point of mutations identified (i.e Cys218Arg) and the classification of the 80 participants (EOAD+, EOAD-, etc). In addition, the method used (whole exome, targeted gene panel, etc) to identify these mutations should be mentioned in the method. The disorder caused by CAG repeat in CACNA1A is SCA6 not EA1 as mentioned in Table 1, also mentioned EA2 Line 254.

  • Line 191, toppgene.cchmc.org website is not accessible
  • Line 239, genes should be in italic
  • Line 239, AVED is not a gene, please correct to TTPA
  • Line 245 Supp. Table IV presented in the manuscript before Supp. Table III.
  • Line 349, Supp. Table XII need to be included
  • Line 445 typo please correct

Author Response

I have read with very interest your interesting manuscript titled “Early Onset Ataxia with Comorbid Dystonia: Clinical, Anatomical and Biological Pathway Analysis Expose Shared Pathophysiology”. The manuscript from Sival and colleagues investigated the prevalence and pathogenesis of EOAD+ in 80 EOA-patients. The main novelty of the study is the EOAD+ cohort. The study is well conducted, and well written. Few suggestions for the authors to improve their manuscript.  

1- the power of study is lack; please could you evaluate if your data could be generalized also at other populations?

Answer: Although EOA is characterized as a rare disorder, we were still able to include a considerable number of 80 patients. We checked on generalizability of the data by evaluating underlying biological pathways in EOA and dystonia gene panels. This (in silico determined) pathway- and network- analysis implicated similar underlying mechanisms (pointing out to hampered energy production and network signal transduction) as in the presently studied ataxia cohort. Additionally, our EOAD+-phenotypes revealed anatomical MRI damage within the basal ganglia-pontine-cerebellar network structures, confirming the presence of an anatomical basis for the concurrence of ataxia and dystonia. In this perspective, we suggest that the presented study data could be interpreted as indicative for larger populations, as well. Hopefully, future collaboration with European and world-wide based ataxia databases will elucidate this.
This information is now added to the discussion section, lines 493-501.

2- The authors conclude Line 513-514 that “it might be a rationalistic approach to test EOA patients with a complete, up to date movement disorder panel (including EOA and dystonia gene lists), instead of with a single EOA gene panel”. That is a good point and the discussion could be opened on the need of WGS on this group of patients followed by a complete up-to-date movement disorder gene list analysis. Moreover, Copy number variation analysis would also be interesting in EAOD+ cohort.

Answer: We thank the reviewer for this remark. As suggested, we included this remark in the first paragraph of the discussion, lines 391-393.

3- Table1: It would be useful to include more information about the point of mutations identified (i.e Cys218Arg) and the classification of the 80 participants (EOAD+, EOAD-, etc).

Answer: On request by the reviewer, we provide more information about the gene mutations and the classification of the 80 participants in the (newly added) Supplementary Table I.

In addition, the method used (whole exome, targeted gene panel, etc) to identify these mutations should be mentioned in the method.

Answer: We added this information to page 3 line 124-125.

The disorder caused by CAG repeat in CACNA1A is SCA6 not EA1 as mentioned in Table 1, also mentioned EA2 Line 258.

Answer: We thank the reviewer. This is corrected in Table I and line 258.

Line 191, toppgene.cchmc.org website is not accessible
Answer: the website is https://toppgene.cchmc.org. This is adjusted in the manuscript.

Line 239, genes should be in italic
Answer: This is corrected.

Line 239, AVED is not a gene, please correct to TTPA
Answer: This is corrected.

Line 245 Supp. Table IV presented in the manuscript before Supp. Table III.
Answer: In the present version of the manuscript we corrected the order.

Line 349, Supp. Table XII need to be included
Answer: We thank the reviewer for this remark, we have now included this Supp Table as Suppl. Table XIII.

Line 445 typo please correct
Anwer: This is corrected.

Reviewer 3 Report

Sival and coauthors describe an innovative study with an integrated clinical and genetic/functional design. The paper is clear and well written and investigate the association of ataxia with dystonia from several point of views generating conclusive results and speculative hypothesis for future studies. 

Comments to the authors:

Introduction and Discussion: the paragraphs describing the hypothetical links between the appearence of dystonia and the neurodegenerative processes  ("dystonia could be explained by neurodegenerative processes") are not clear. It seems that dystonia is expected in patients with a long disease duration but literature clearly doesn't support that observation, as early onset patients can also present dystonia. Please discuss this issue and clarify in the Introduction/Discussion

Table1: if possible, please list the patients on the basis of onset or gene name, to facilitate the reader

MRI Methods/Results: please specify what kind of "damage/abnormalities" was observed at brain MRI. If possible list the different types of "abnormalities" in the Results and Supplementary table 1

Author Response

Introduction and Discussion: the paragraphs describing the hypothetical links between the appearance of dystonia and the neurodegenerative processes ("dystonia could be explained by neurodegenerative processes") are not clear. It seems that dystonia is expected in patients with a long disease duration but literature clearly doesn't support that observation, as early onset patients can also present dystonia. Please discuss this issue and clarify in the Introduction/Discussion

Answer: We agree with the reviewer that neurodegenerative processes are not the cause for comorbid dystonia in EOA. However, this has been suggested in the previous literature of SCA’s (reference 5) and therefore, analogously, we included this point as one of the potential, hypothetical causes that could be addressed. In the present version of the manuscript, we address more clearly that we consider this hypothetical cause as an unlikely explanation for dystonic comorbidity in EOA. See introduction section lines 64-66 and discussion section lines 377-380.

Table1: if possible, please list the patients on the basis of onset or gene name, to facilitate the reader

Answer: For encoding reasons, leading back to the source documents, we strongly prefer to leave the presented order of patients intact. As this does not change anything to the study results, study contents and/or interpretation of the data, we hope that this can be accepted.

MRI Methods/Results: please specify what kind of "damage/abnormalities" was observed at brain MRI. If possible list the different types of "abnormalities" in the Results and Supplementary Table I

Answer: We now provide this information in Supplementary Table II (previous Suppl Table I, in the old version). We also included this information in the results.